# 1,5-Anhydro-D-Fructose Exhibits Satiety Effects via the Activation of Oxytocin Neurons in the Paraventricular Nucleus

**DOI:** 10.3390/ijms24098248

**Published:** 2023-05-04

**Authors:** Masanori Nakata, Yuto Yamaguchi, Hikaru Monnkawa, Midori Takahashi, Boyang Zhang, Putra Santoso, Toshihiko Yada, Ikuro Maruyama

**Affiliations:** 1Department of Physiology, School of Medicine, Wakayama Medical University, Kimiidare 811-1, Wakayama 641-8509, Japan; d2066093@wakayama-med.ac.jp (Y.Y.); d2066092@wakayama-med.ac.jp (H.M.); d1766047@wakayama-med.ac.jp (M.T.); zhangby@wakayama-med.ac.jp (B.Z.); 2Department of Physiology, Division of Integrative Physiology, School of Medicine, Jichi Medical University, Shimotsuke 329-0498, Japan; santosounand@gmail.com; 3Center for Integrative Physiology, Kansai Electric Power Medical Research Institute, Kyoto 604-8436, Japan; toshihiko.yada@kepmri.org; 4Department of Systems Biology in Thromboregulation, Graduate School of Medical and Dental Science, Kagoshima University, Kagoshima 890-8520, Japan; maruyama-i@eva.hi-ho.ne.jp

**Keywords:** 1,5-Anhydro-D-fructose, oxytocin, feeding behavior, PVN

## Abstract

1,5-Anhydro-D-fructose (1,5-AF) is a bioactive monosaccharide that is produced by the glycogenolysis in mammalians and is metabolized to 1,5-anhydro-D-glucitol (1,5-AG). 1,5-AG is used as a marker of glycemic control in diabetes patients. 1,5-AF has a variety of physiological activities, but its effects on energy metabolism, including feeding behavior, are unclarified. The present study examined whether 1,5-AF possesses the effect of satiety. Peroral administration of 1,5-AF, and not of 1,5-AG, suppressed daily food intake. Intracerebroventricular (ICV) administration of 1,5-AF also suppressed feeding. To investigate the neurons targeted by 1,5-AF, we investigated c-Fos expression in the hypothalamus and brain stem. ICV injection of 1,5-AF significantly increased c-Fos positive oxytocin neurons and mRNA expression of oxytocin in the paraventricular nucleus (PVN). Moreover, 1,5-AF increased cytosolic Ca^2+^ concentration of oxytocin neurons in the PVN. Furthermore, the satiety effect of 1,5-AF was abolished in oxytocin knockout mice. These findings reveal that 1,5-AF activates PVN oxytocin neurons to suppress feeding, indicating its potential as the energy storage monitoring messenger to the hypothalamus for integrative regulation of energy metabolism.

## 1. Introduction

The most common metabolic disorder, type 2 diabetes mellitus (T2D) is characterized by chronic hyperglycemia that is caused by insulin resistance and/or deficient insulin release. Glycogen is a polymer of glucose stored in the liver that is catabolized by glycogen phosphorylase to supply glucose to the blood during fasting and exercise. Liver glycogen levels fluctuate with diet and circadian rhythms, but their regulation is impaired in T2D patients with insulin resistance, resulting in increased glucose output [1]. The bioactive monosaccharide, 1,5-Anhydro-D-fructose (1,5-AF) is also metabolized from glycogen stored in the liver by the action of α-1,4-lyase. 1,5-AF is further reduced to 1,5-Anhydro-D-glucitol (1,5-AG) by an NADPH-dependent reductase [2,3,4]. 1,5-AG has already been clinically applied as an indicator of glycemic control and reduced in T2D patients [5]. Both 1,5-AF and 1,5-AG are found in mammalian tissues, algae, fungi, bacteria and plants [6,7,8], and are also ingested by diet. 1,5-AF has antimicrobial, antioxidant and anti-inflammatory properties [9], but its anti-obesity effects have not been clarified.

Feeding behavior, one of the most important activities for energy homeostasis, is regulated by a neural network of particular neurons located in the arcuate nucleus (ARC), the ventromedial nucleus (VMH), the dorsomedial nucleus (DMH), the lateral hypothalamic area (LHA) and the paraventricular nucleus (PVN) in the hypothalamus. The neurons in ARC sense peripheral metabolic signal molecules, such as leptin, ghrelin, and glucose [10]. The neurons in ARC project widely in the hypothalamus, but projections to the paraventricular nucleus are particularly important for energy metabolism via the endocrine and autonomic systems. The neurons of the PVN express numerous neuropeptides [11]. Neuronal populations within the PVN expressing anorexigenic neuropeptides such as oxytocin, corticotropin-releasing hormone (CRH), arginine vasopressin (AVP) and nesfatin-1/nucleobindin-2 (NUCB2) are critical for the regulation of energy metabolism [12,13]. Dysfunction of this neural network is the cause of obesity, and intervention with functional molecules is useful in the treatment of obesity.

For half a century, it has been clear that 1,5-AG is present in cerebrospinal fluid [14]. Furthermore, the metabolic pathway from glycogen to 1,5-AF and 1,5-AG in the central nervous system has also been clarified [3]. However, the physiological functions of 1,5-AF and 1,5-AG in central nervous system, especially regarding energy metabolism, remain unclear. Thus, in this study, we analyzed the effects of 1,5-AF on feeding behavior and identified its target neurons.

## 2. Results

In this study, peroral administration of 1,5-AF at 0.1 mg decreased cumulative food intake for 24 h after injection in C57BL/6J mice (Figure 1A). 1,5-AF at 0.01 mg did not alter cumulative food intake for 24 h, showing a dose-dependent effect. Since 1,5-AF is quickly metabolized to 1,5-AG in vivo, an investigation was conducted to compare the effects of 1,5-AF and 1,5-AG on feeding behavior. Interestingly, the administration of 1,5-AF at 0.1 mg significantly reduced cumulative food intake compared to the administration of 1,5-AG at 0.1 mg (Figure 1A). Thus, oral administration of 1,5-AF possessed a distinctive satiety effect compared to 1,5-AG.

To explore whether the effect of 1,5-AF on feeding is mediated via the central nervous system, we examined the effect of ICV administration of 1,5-AF on feeding behavior. Considering osmotic effects, ICV administration experiments were conducted using 1,5-AG as a control. The ICV injection of 0.3 µg 1,5-AF, compared to same dose of 1,5-AG, significantly reduced cumulative food intake for 24 h. after injection (Figure 1C). The feeding inhibitory effects of 1,5-AF were more potent when administered intracerebroventricularly compared to peroral administration. These results suggest that the central nervous system is involved in the anorexigenic effect of 1,5-AF feeding inhibition.

To clarify the neural nuclei in which 1,5-AF acts, we investigated c-Fos expression, an indicator of the activated neurons, in the areas of the hypothalamus and brain stem implicated in the regulation of feeding behavior. ICV administration of 1,5-AF significantly increased the number of c-Fos expression neurons in the PVN and nucleus tractus solitarius (NTS) (Figure 2A,B). Comparing these two nerve nuclei, the increment of c-Fos expression was exceptionally high (four-fold) in PVN (Figure 2B), suggesting that the neurons in the PVN play a role in the satiety effect of 1,5-AF.

Next, to identify the subpopulation of PVN neurons activated by 1,5-AF, the expression of neuropeptides in the PVN induced by ICV administration of 1,5-AF was investigated using qPCR. Administration of 1,5-AF increased mRNA expression of *Oxt* while *Avp*, *Crh* and *Nucb2* mRNA expression was not altered (Figure 3A). Moreover, ICV administration of 1,5-AF increased the plasma oxytocin concentration compared to 1,5-AG (Figure 3B). Furthermore, ICV administration of 1,5-AF significantly increased c-Fos positive oxytocin neurons only in the PVN, not in the SON (Figure 3C–G). These results suggest that ICV administration of 1,5-AF activates oxytocin neurons in PVN.

To investigate whether 1,5-AF directly activates oxytocin neurons in PVN, we measured cytosolic Ca^2+^ concentration ([Ca^2+^]_i_) in the isolated single PVN neurons. 1,5-AF at 10 µg/mL increased [Ca^2+^]_i_ in the PVN neurons that were subsequently shown to be immunoreactive to oxytocin (Figure 4A). Twelve of 52 oxytocin neurons (23%) responded to 1,5-AF at 100 µg/mL (Figure 4B). These results indicated that 1,5-AF is the direct regulator of oxytocin neurons. On the other hand, about 17% of non-oxytocin neurons were also activated by 1,5-AF, suggesting the presence of other candidate neurons besides oxytocin.

To confirm oxytocin mediates the anorexigenic action of 1,5-AF, we investigated the satiety effect of 1,5-AF in oxytocin knockout (KO) mice. In KO mice, peroral administration of 1,5-AF failed to reduce cumulative food intake for 24 h (Figure 5A). In addition, peroral administration of 1,5-AF suppresses food intake in littermate wild-type mice compared to KO mice (Figure 5B). Thus, to clarify whether orally administered 1,5-AF activates oxytocin neurons in the PVN, we investigated the expression of oxytocin in the PVN of wild type mice. The peroral administration of 1,5-AF at 0.1 mg potently increased mRNA expression of Oxytocin (Figure 5C). It has been previously reported that 1,5-AF enhances the secretion of the anorexigenic hormone, GLP-1 [15]. However, 1,5-AF did not increase plasma concentrations of GLP-1 in this study (Figure 5D). These results, taken together, suggest that the oxytocin neurons in PVN mediate the anorexigenic effect of 1,5-AF. 

## 3. Discussion

In this study, we showed that peroral and ICV administration of 1,5-AF reduces the daily food intake of wild mice. ICV administration injection of 1,5-AF significantly increased c-Fos and oxytocin expression in the PVN. Moreover, 1,5-AF directly activates isolated oxytocin neurons in PVN. Furthermore, the satiety effect of 1,5-AF was abolished in oxytocin knockout mice. These results suggest that 1,5-AF exhibits satiety effects via the activation of oxytocin neurons in the PVN. 

1,5-AF, 1,5-AG and glycogen are widely distributed in various organs in the body [3]. The concentration of 1,5-AF in the brain has been estimated to be approximately ≈0.25 µM (40 µg/L) [16]. Thus, the dose of AF administered intracerebroventricularly in this study is a high concentration, perhaps a pharmacological dose, but it is 1/300th of the dose administered perorally. By radioisotope tracing experiments, 1,5-AF has been shown to be efficiently absorbed from the gastrointestinal tract and excreted in the urine [17]. 1,5-AG is transported by D-fructose and D-mannose selective transporters, and sodium glucose transporter 4 (SGLT4/SCL5A9) [18,19]. SGLT4 is strongly expressed in the intestinal tract and in the kidney, and 1,5-AF is presumed to be absorbed via SGLT4 alongside 1,5-AG. Furthermore, SGLT4 is also localized in the brain, and thus it is also conceivable that a portion of orally administered 1,5-AF is transported to and acts in the brain [16].

Neuronal systems involved in the regulation of energy intake sense and integrate inputs from periphery energy metabolism. These signals inform the brain of energy storage and availability in the body. The afferent signals that are crucial for the regulation of energy intake include nutrient-related signals such as, glucose, amino acids and fatty acids. Within the hypothalamus, the ARC and VMH are the first order nucleuses of feeding control, largely projected to the PVN, which is the control center for homeostatic, neuroendocrine, and autonomic function [11]. In the present study, 1,5-AF activated oxytocin neurons and anorexigenic neurons in the PVN within the hypothalamus. Oxytocin neurons in the PVN play an important role in feeding regulation by the reward system in addition to feeding regulation by the homeostatic system [20]. The activating effects of 1,5-AF on oxytocin neurons suggest therapeutic applications not only for feeding suppression but also for addiction and schizophrenia. 

Previously, we have shown that NUCB2/Nesfatin-1 in the PVN is an activator of oxytocin neurons [12,13]. However, 1,5-AF did not alter the expression of NUCB2, indicating that it activates oxytocin in a NUCB2/Nesfatin-1-independent manner. On the other hand, the presence of neurons outside of oxytocin neurons as targets of 1,5-AF were found (Figure 4B). Further study of this neuron is expected to lead to the elucidation of new oxytocin neuron activity pathways and the regulation of feeding behavior. 

1,5-AF is detected in several organs, including the brain, but not in plasma. These results suggest that 1,5-AF is produced by glycogen metabolism in each organ [3,21]. Especially astrocytes storage glycogen in the brain and may derive 1,5-AF to neurons. It is well known that astrocytes play a dynamic role in the regulation of oxytocin release by the hypothalamo-neurohypophysial system [22]. Under basal conditions, astrocytes attenuated the neuronal excitability. After physiological activation, such as lactation, morphological changes of astrocytes actively facilitate neuron-neuron interactions. Thereby, new synapses are formed and excitatory transmitter action is prolonged. On the other hand, receptors for oxytocin also are expressed on hypothalamic astrocytes in primary cultures. Prader-Willi syndrome is a neurodevelopmental disorder complicating hyperphagia and obesity. Previous reports implicated dysfunctional signaling of oxytocin as one of the pathological mechanisms in Prader-Willi syndrome, and intranasal oxytocin administration produced effective therapeutic results in humans [23,24]. In the Magel2-deficient mouse, mouse models of Prader-Willi syndrome, the number of oxytocin neurons and the oxytocin receptor-expressing astrocytes were reduced in the PVN [25]. These findings suggest that neuro-glial interactions in the PVN are momentous for oxytocin neuron activity and that 1,5-AF may be involved in these interactions, and further studies are expected.

It has been previously reported that 1,5-AF promotes a feeding suppressor, GLP-1 secretion [15,26], but peroral administration of 1,5-AF at 0.1 mg did not alter GLP-1 secretion in this study. In previous reports, 150 mg of AF have been administered, which is approximately 1500 times the dose used in this study [15]. Furthermore, GLP-1 secretory effects have been observed with simultaneous administration of 1,5-AF and glucose in previous reports, not with peroral administration of 1,5-AF alone. It is believed that orally administered glucose itself has GLP-1 secretion-promoting effects [27], and 1,5-AF may potentiate these effects. In the present study, 1,5-AF was administered during fasting at the end of the light period, and although it is possible that the facilitatory effect was not observed, we estimate that the inhibition of feeding by 1,5-AF is GLP-1-independent in this study.

In conclusion, our results indicated that exogenous 1,5-AF has an inhibitory effect on feeding and one of its mechanisms activates oxytocin neurons in the PVN. Endogenous 1,5-AF may be derived from astrocytes and modulate the excitability of oxytocin neurons in the PVN. Future studies on the neuro-glial interaction mediated by 1,5-AF would provide additional insights into energy homeostasis, which may lead to novel treatments for obesity.

## 4. Materials and Methods

### 4.1. Animals

Male C57BL/6 mice aged 8–12 weeks were obtained from Japan SLC (SLC, Hamamatsu, Japan). The Oxytocin KO C57BL/6J mice were donated by Dr. K. Nishimori [28]. All KO mice and littermates were genotyped by PCR of genomic DNA isolated from tail tips. The animals were housed individually under a 12-h light/dark cycle condition (8:00 light on). All animals in this study were fed regular chow food (CE-2; CLEA, Osaka, Japan) containing 24.9% protein, 4.6% fat, and 4.1% fiber. Food and water were available ad libitum except during particular experiments. Mice were sufficiently habituated to handling before experiments. All experiments were conducted in accordance with the guidelines of the Jichi Medical University Animal Care and Use Committee (Approval number: 17176-01) and Wakayama Medical University Animal Care and Use Committee (Approval number: 1052 and 2020-50). The study was carried out in compliance with the ARRIVE guidelines.

### 4.2. Measurement of Food Intake

The mice were randomly divided into four treatment groups (n = 6 in each group). For feeding experiments, 1,5-AF and 1,5-AG (Sanasu, Kagoshima, Japan) were dissolved in distilled water at 1 mg/mL. Following deprivation of food for 2 h before the dark phase, the animals were perorally administrated 100 μL of distilled water (placebo control), 1,5-AF solution and 1,5-AG solution at 30 min using a stainless-steel feeding needle (KN-348; Natsume, Tokyo, Japan) before the dark phase (19:30). Cumulative food intake was measured for the following 24 h. 

Oxytocin KO mice and wild-type littermates (n = 5 in each group) were housed in individual cages and habituated to peroral administration for 1 week before experiments. For food intake measurements, mice were administered distilled water on day 1, 1,5-AF (0.1 mg) on day 2, and distilled water on day 3 at 30 min before the dark phase (19:30). Cumulative food intake was measured for the following 24 h.

### 4.3. Intracerebroventricular (ICV) Cannula Implantation and ICV Injection

C57BL/6 mice aged 9 weeks underwent surgery to implant the guide cannula (ICM-23G09; Intermedial, Osaka, Japan) into lateral ventricles using a stereotaxic instrument (Kopf Instruments, Tujunga, CA, USA). The cannula tip was located at 0.5 mm caudal to the bregma, 1.0 mm lateral to the midline and 2.2 mm below the skull. Mice were allowed to recover from surgery for one week and habituated to ICV injection before experiments. 

For ICV injection experiments, 1,5-AF and 1,5-AG were dissolved in sterile saline at 0.1 μg/μL and ICV injected at 0.3 μg/3 μL using a micro injector syringe (1705 N, Hamilton, OH, USA) attached to polyethylene tubing with a 30-gauge needle. After 2 h fasting prior to injection, mice were injected and refed immediately upon injection. The agents were injected 30 min before the onset of the dark cycle (19:30). Cumulative food intake was measured for the following 24 h. After experiments, cannula placement was confirmed by the appearance of dye in the lateral ventricular system.

### 4.4. c-Fos Expression

c-Fos expression was examined as reported in Santoso et al. [29]. Briefly, at 2 h. after ICV injection of 1,5-AF or 1,5-AG, animals were perfused transcardially with heparinized saline followed by 4% paraformaldehyde (PFA) in 0.1 M phosphate buffer for 20 min. The brains were removed and subjected to immunohistochemistry. Coronal sections were cut at 40 µm thickness using a freezing microtome and collected in phosphate buffered saline (PBS). Brain sections were pretreated with 0.3% H_2_O_2_ for 20 min to inhibit endogenous peroxidases. After rinsing, sections were incubated with 2% bovine serum albumin and 2% normal goat serum for 30 min and with rabbit anti-c-Fos antibody (sc-52, Santa Cruz, 1:5000) overnight at 4 °C. After washing, the sections were incubated with biotinylated goat anti-rabbit IgG for 30 min and sequentially incubated with avidin-biotin complex (ABC) reagent for 30 min (Vector Laboratories, Burlingame, CA, USA). Color development was performed using a nickel-diaminobenzidine (DAB) solution (0.3% nickel ammonium sulfate, 0.02% DAB, and 0.005% H_2_O_2_ in 0.05 M Tris buffer) for 5 min. Images were acquired with Olympus BX51 microscope (Olympus, Tokyo, Japan).

For analyses of c-Fos expression, the number of neurons with c-Fos positive nuclei was counted for every section of every mouse. The average number of neurons with nuclei positive for c-Fos per section was calculated for each mouse.

### 4.5. Immunohistochemistry for c-Fos and Oxytocin

Double staining of c-Fos and oxytocin was examined as reported in Nakata et al. [30]. Briefly, at 2 h. after ICV injection of 1,5-AF or 1,5-AG, animals were fixed with 4% PFA in 0.1 M phosphate buffer. The brains were removed and subjected to immunohistochemistry. The coronal sections with 40 µm thickness were cut using a freezing microtome. After washing in PBS, endogenous peroxidases in sections were blocked by incubation in 0.3% H_2_O_2_ for 30 min. After washing, blocking was performed with 2% normal goat serum and 2% bovine serum albumin for 60 min. Next, sections were incubated with primary rabbit anti-c-Fos antibodies (sc-52, Santa Cruz, 1:5000) overnight at 4 °C. After washing, the sections were incubated with biotinylated goat anti-rabbit IgG (Vector Laboratories; 1:1000) for 30 min and incubated with an avidin-biotin complex (ABC) reagent for 30 min (Vector Laboratories). Color development was performed using a nickel-diaminobenzidine (DAB) solution (0.3% nickel ammonium sulfate, 0.02% DAB, and 0.005% H_2_O_2_ in 0.05 M Tris buffer) for 5 min. After washing in PBS, sections were treated with an avidin and biotin blocking solution (Vector Laboratories) and then incubated with primary mouse anti-oxytocin antibodies (MAB5296; Millipore 1:1000) overnight at 4 °C. After washing, sections were incubated with biotinylated horse anti-mouse IgG antibodies for 30 min and incubated in an ABC reagent for 30 min. After rinsing in PBS and Tris buffers, color development was performed using a DAB solution (0.02% DAB and 0.005% H_2_O_2_ in Tris buffer). Lastly, sections were mounted on slides, and cover-slipped with Entellan™ new rapid mounting medium (Merck, Darmstadt, Germany).

For analyses of c-Fos expression in oxytocin neurons, the number of neurons co-positive for oxytocin and/or c-Fos were counted for every section of every mouse. The average number of oxytocin neurons with nuclei positive for c-Fos per section was calculated for each mouse. 

### 4.6. qPCR

qPCR was performed as reported in Zhang et al. [31]. Briefly, at 3 h. after ICV or P.O administration of agents, bilateral PVN areas were resected from brain slices containing hypothalamuses from C57BL/6 mice. Total RNA was isolated by TRIzol (Invitrogen, Carlsbad, CA, USA) and degraded contaminated genomic DNA using RQ1-DNase (Promega, Madison, WI, USA). First-strand cDNA syntheses were prepared using PrimeScript™ RT Master Mix (Takara bio, Shiga, Japan). The qPCR was performed using SYBR Premix Ex Taq II polymerase in Thermal Cycler Dice in a Thermal Cycler Dice Real Time System (Takara Bio, Tokyo, Japan). Expression levels of mRNAs were normalized to that of glyceraldehyde-3-phosphate dehydrogenase (*Gapdh*) by the ΔΔCT method. Sequences of primers for neuropeptides and *Gapdh* are listed in Table 1. 

### 4.7. Measurement of Cytosolic Calcium Concentration ([Ca^2+^]_i_)

Measurement of [Ca^2+^]_i_ was also performed as described in Nakata et al. [30]. C57BL/6J mice aged 5–6 weeks were used for measurements of [Ca^2+^]_i_. After anesthesia, the brain was removed immediately and placed in iced Krebs-Ringer bicarbonate buffer (KRB) solution (129 mM NaCl, 5.0 mM NaHCO_3_, 4.7 mM KCl, 1.2 mM KH_2_PO_4_, 2.0 mM CaCl_2_, 1.2 mM MgSO_4_ and 10 mM HEPES at pH 7.4) containing 5.6 mM glucose. The brain slice containing the PVN was coronally sectioned and a block of tissue containing both sides of the PVN was dissected. After washing in cold KRB, the dissected block was incubated in the KRB supplemented with 20 U/mL papain, 0.015 mg/mL DNase, 0.75 mg/mL BSA, and 1 mM cysteine for 15–16 min at 36 °C in a water bath shaking 40 times per minute. Then, single neurons were placed on a cover glass and incubated at 35 °C for 30 min. [Ca^2+^]_i_ was measured by ratiometric fura-2 fluorescence imaging. After enzyme treatment, the tissues were triturated gently with custom made glass pipettes to disperse into single neurons. Briefly, isolated single neurons on coverslips were loaded with 2 µmol/l fura-2/Acetoxymethyl ester (Dojin Chemical, Kumamoto, Japan) for 30 min in a 35 °C incubator. Next, the coverslips were mounted in a chamber, and superfused with reagents and HKRB containing 5.6 mM glucose at 1 mL/min. Neurons were excited at 340 and 380 nm alternately every 5 s. Fluorescence emission at 510 nm (F340 and F380, respectively) was acquired and ratio (F340/F380) images were produced by the Aquacosmos image processing system (Hamamatsu Photonics, Hamamatsu, Japan). Ratio amplitudes of [Ca^2+^]_i_ increases in response to 1,5-AF were calculated by subtracting pre-stimulatory [Ca^2+^]_i_ levels from peak [Ca^2+^]_i_ levels during the addition of 1,5-AF. Increases in [Ca^2+^]_i_ within 10 min of the addition of 1,5-AF with a ratio amplitude of 0.05 or greater were considered positive responses. Analyses were performed using live cells that responded to glutamate (10^−5^ M). The single neurons analyzed for [Ca^2+^]_i_ measurements were subsequently immune-stained for oxytocin. The cells were fixed with 4% PFA overnight. Cells were rinsed with PBS for 5 min three times and pretreated with 0.3% H_2_O_2_ in PBS for 10 min. After blocking in 2% normal goat serum and 2% bovine serum albumin in PBS for 30 min, cells were incubated with primary rabbit anti-oxytocin monoclonal antibodies (EPR20973; Abcam, Cambridge, 1:1000). The primary antibody was then washed with PBS and incubated with biotinylated secondary antibody raised against rabbit IgG (Vector Laboratories; 1:1000) for 30 min at room temperature. The secondary antibody was then rinsed with PBS for 5 min three times, and the cells were labeled with ABC complex (Vector Laboratories) for 30 min. Finally, color was developed with DAB. Immunohistochemical staining using this anti-oxytocin monoclonal antibody revealed positive neurons specifically in the PVN and SON of the mouse brain. (Appendix A).

### 4.8. Measurement of Plasma Oxytocin and GLP-1 Concentration

Briefly, at 1 h after ICV administration of agents, blood was collected in tubes containing heparin (final concentration; 50 IU/mL), and aprotinin (final concentration; 500 kIU/mL) and separated by centrifugation for 10 min at 3000× *g* at 4 °C. Plasma concentrations of oxytocin were measured with an Oxytocin ELISA kit (Wako, Osaka, Japan). 

The blood was collected from the portal vein of anesthetized mice at 1 h after peroral administration of 1,5-AF (0.1 mg/100 μL) or distilled water, respectively. The sampling syringe contained heparin, aprotinin and DPP-4 inhibitor (Merck). Plasma was separated by centrifugation for 10 min at 3000× *g* at 4 °C. Plasma concentrations of GLP-1 were measured with a GLP-1 ELISA kit (Fujifilm WAKO, Osaka, Japan).

### 4.9. Statistical Analysis

Data were presented as mean ± SEM. The one-way ANOVA was applied for two groups. Post-hoc multiple comparison was generated using Bonferroni test. *p* < 0.05 was considered significant. For feeding experiment of daily administration, the two-way ANOVA with Bonferroni’s multiple comparison test was applied.

## Figures and Tables

**Figure 1 ijms-24-08248-f001:**
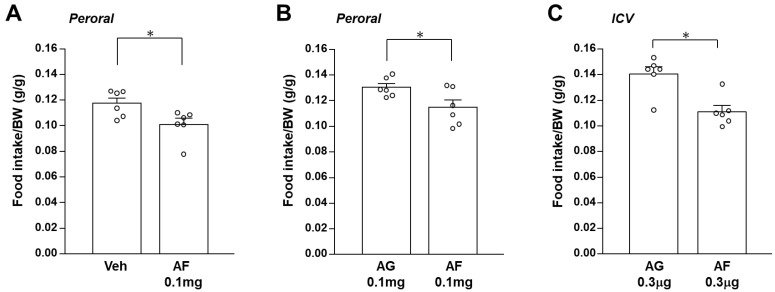
Peroral and ICV administration of 1,5-AF reduces cumulative food intake. (**A**) Cumulative food intake per body weight measured for 24 h. after peroral administration of 1,5-AF (0.1 mg/100 μL) or distilled water. (n = 6 for each group). (**B**,**C**) The effect of 1,5-AF (black bars) or 1,5-AG (white bars) on cumulative food intake measured for 24 h. after peroral administration (**B**) and ICV administration (**C**). (n = 6 for each group). Data are presented as mean ± SEM. * *p* < 0.05.

**Figure 2 ijms-24-08248-f002:**
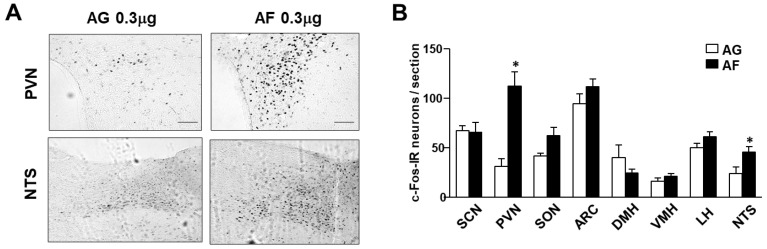
ICV injection of 1,5-AF increases c-Fos expression in the PVN. (**A**) Immunohistochemical staining of c-Fos in the PVN and NTS after injection of 1,5-AF (right panels) or 1,5-AG (left panels). (**B**) Number of c-Fos-immunoreactive (IR) neurons per section in the hypothalamic and brain stem regions. SCN; Suprachiasmatic nucleus, PVN; Paraventricular nucleus, SON; Supraoptic nucleus, ARC; Arcuate nucleus, DMH; dorsomedial nucleus in hypothalamus, VMH; Ventromedial nucleus in hypothalamus, LH; lateral hypothalamic area, NTS; nucleus tractus solitarius. Data are presented as mean ± SEM. * *p* < 0.05. n = 5–6. Scale bars represent 200 μm.

**Figure 3 ijms-24-08248-f003:**
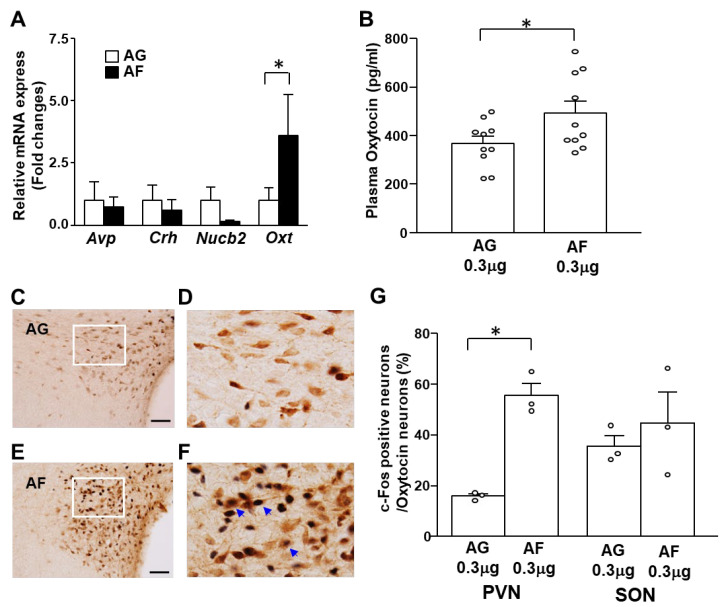
1,5-AF activates PVN oxytocin neuron in vivo. (**A**) Relative mRNA expression (fold change) of neuropeptides in PVN after ICV injection of 1,5-AF (0.3 μg/3 μL) (black bars) or 1,5-AG (0.3 μg/3 μL) (white bars) at 2 h. (n = 6 for each group). (**B**) Plasma oxytocin concentrations after ICV injection of 1,5-AF (0.3 μg/3 μL) (black bars) or 1,5-AG (0.3 μg/3 μL) (white bars) at 2 h. (n = 10 for each group). (**C**–**F**) Double immunostaining of c-Fos and oxytocin in the PVN after ICV administration of 1,5-AG (**C**,**D**) and 1,5-AF (**E,F**). (**D**,**F**) indicate the enlarged image of the white square area in (**C**,**F**). (**G**) Incidence of c-Fos positive neurons in oxytocin neurons in the PVN and SON. Blue arrows indicate the neurons IR to both c-Fos and oxytocin. Scale bar: 200 μm. Data are presented as mean ± SEM; * *p* < 0.05.

**Figure 4 ijms-24-08248-f004:**
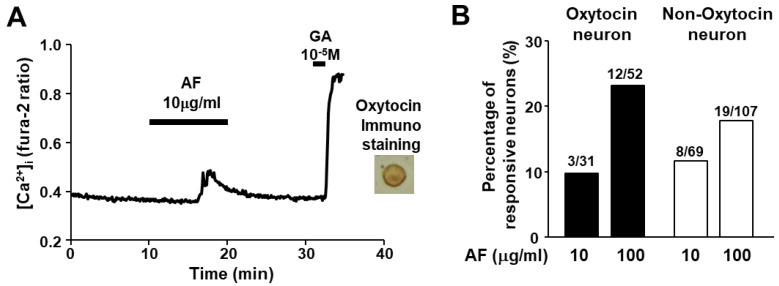
1,5-AF increases [Ca^2+^]_i_ in single oxytocin neurons in the PVN. (**A**) 1,5-AF at 100 μg/mL increased [Ca^2+^]_i_ (left panel) in a single PVN neuron under perfusion of KRB containing 5.6 mmol/l glucose. This neuron also responded to 10^−5^ M glutamic acid (GA) and was shown to be IR to oxytocin by subsequently immune-staining. [Ca^2+^]_i_ is expressed as fluorescence fura-2 ratio (F340/F380). (**B**) Responsiveness of [Ca^2+^]_i_ responses to 1,5-AF in oxytocin neurons and non-oxytocin neurons were indicated by percentage. The number above the bar indicates the number of neurons that responded over that examined. Data are presented as mean ± SEM.

**Figure 5 ijms-24-08248-f005:**
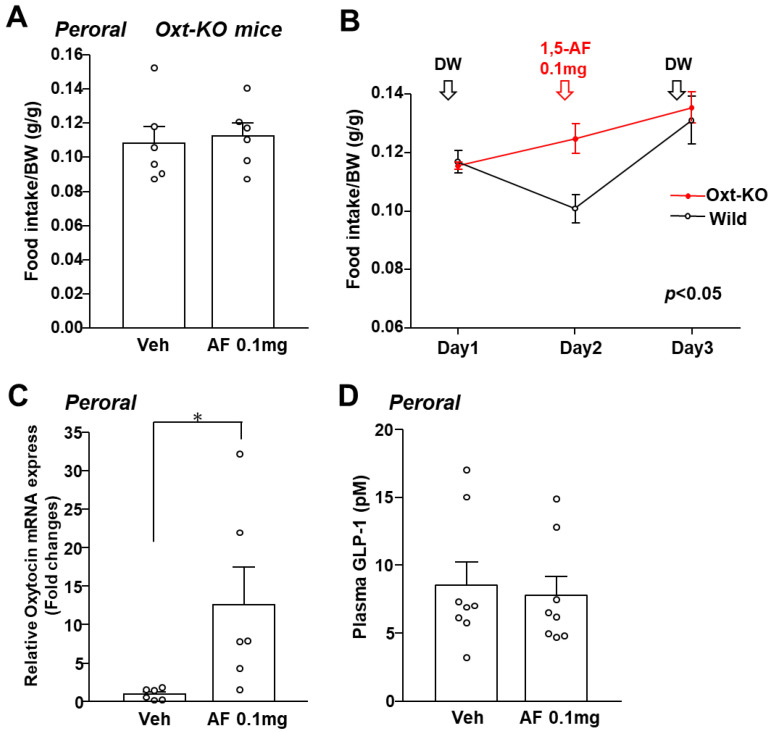
Deficiency of oxytocin abolishes the anorexigenic effect of 1,5-AF. (**A**) Cumulative food intake in oxytocin knock-out (Oxt-KO) mice for 24 h after peroral administration of 1,5-AF (0.1 mg per mouse) or distal water. (n = 6 for each group). (**B**) Oxt-KO mice (red) and wild-type littermates (black) were administrated distilled water (100 mL) on day 1, 1,5-AF (0.1 mg/100 mL) on day 2, and distilled water (100 mL) on day 3 at 30 min before the dark phase (19:30). Cumulative food intake was measured for the following 24 h. (n = 5 for each group) (**C**) Relative mRNA expression (fold change) of oxytocin in the PVN of C57B/6 mice after peroral administration of 1,5-AF (0.1 mg per mouse) or distal water (white bars) at 2 h. (n = 6 for each group). (**D**) Plasma GLP-1 concentrations of C57B/6 mice after peroral administration of 1,5-AF (0.1 mg per mouse) or distal water (white bars) at 1 h. (n = 8 for each group). Data are presented as mean ± SEM. * *p* < 0.05.

**Table 1 ijms-24-08248-t001:** Primers used in the RT-PCR experiment.

Gene	Forward	Reverse
*Avp*	5′-CATCTCTGACATGGAGCTGAGA-3′	5′-GGCAGGTAGTTCTCCTCCTG-3′
*Crh*	5′-TCTCTCTGGATCTCACCTTCCACC-3′	5′-AGCTTGCTGAGCTAACTGCTCTGC-3′
*Nucb2*	5′-GTCACAAAGTGAGGACGAGACTG-3′	5′-TGGTTCAGGTGTTCAAACTGCTTC-3′
*Oxt*	5′-TGTGCTGGACCTGGATATGCGCA-3′	5′-GGCAGGTAGTTCTCCTCCTG-3′
*Gapdh*	5′-GGCACAGTCAAGGCTGAGAATG-3′	5′-ATGGTGGTGAAGACGCCAGTA-3′

## Data Availability

The data used to support the findings of this study is available from the corresponding authors upon reasonable request.

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
