# Peer review of "1,5-Anhydro-D-Fructose Exhibits Satiety Effects via the Activation of Oxytocin Neurons in the Paraventricular Nucleus"

_ijms, 2023, doi:10.3390/ijms24098248_

Round 1
Reviewer 1 Report
Nakata et al. investigated the role of 1,5-AF in the effect of satiety and also the brain region targeted by 1,5 AF. The major finding of this study was that peroral and ICV administration of 1,5-AF reduced cumulative food intake. Additional findings in this study were that ICV injection of 1,5 AF increased c-Fos positive neurons in the paraventricular hypothalamus, 1,5 AF activated PVN oxytocin neurons and also increased cytosolic Ca2+ in these neurons. Based on these findings, the authors suggested that 1,5 AF have an inhibitory effect on feeding and the pathway involved in this process leads to the activation of oxytocin neurons in PVN. However, it is possible that other neurons in the PVN or other parts of the brain also contribute towards the observed satiety effects.
This study is a starting point to elucidate possible effects of 1,5 AF in regulating appetite, it is very concise and informative but I have a few additional questions and clarifications regarding experimental and analysis methodology.
Major comments:
-
Dose for Intracerebroventricular Injection - explain why the high dose was chosen for ICV injection. How was this dose chosen? Is it a common dose used in other protocols? Why and how was the high dose for ICV chosen? Is there a satiety effect at lower doses?
-
Figure 2 - How was c-Fos quantification performed? Please specify the details, the manner it was performed and how many people quantified it.
-
Some aspects of Figure 3 need to be clarified:
-
Figure 3D - relabel y-axis as “number of responsive neurons” or use another clearer label
-
In the methods section: “When [Ca2+] changed within 10 min after addition of agents and their ratio amplitude of 0.2 or greater was considered positive responses”.
-
Define what ratio amplitude means.
-
Explain how the threshold of 0.2 was set and the reason for the setting.
-
In Figure 3C, the amplitude difference does not look like 0.2. Was it counted as an active neuron?
-
Explain the use of glutamate in the text or methods. as it meant to be a positive control?
-
For Figure 3C-D, I assume that the authors also found neurons that were not oxytocin positive. Can they provide the data related to the incidence of response in non-oxytocin neurons? How does this incidence rate compare to the oxytocin neurons?
-
Figure 4: The data for the Oxt KO mutants requires a positive control (ideally sibling WT mice or age-matched mice with the same genetic background, with experiments performed at the same time). It cannot simply be compared to Figure 1.
-
Some claims in the study need to be toned down or more supporting evidence added with new data (e.g. Line 174: “In the present study, 1,5-AF specifically activated oxytocin neurons in the PVN within the hypothalamus”). The specificity to oxytocin is not that well-established from the current set of experiments, for example, in Figure 2A, there is no evidence that the activated neurons are OXT positive (double fluorescent immunostaining would be necessary to answer this question), and in Figure 3B, no other other plasma hormones were analyzed. Further, in Figure 3C-3D, there is no mention of the response rate of OXT negative neurons. The OXT KO experiments are not controlled and hence not convincing. Overall, if the authors cannot provide additional data, they will at least need to tone down these mechanistic claims of OXT involvement, which I actually do not think will take away from the impact of this study.
-
Statistical analysis - I would appreciate it if the authors could add to their plots the individual data points, not just the average and SEM. Overall, effect sizes seem small, would be good to note/discuss this in the text.
Minor comments:
In general, there are a number of typos and aspects that could be clarified to make the paper easier to read. Some examples (non-exhaustive) below:
-
Explain briefly what c-Fos is.
-
Line 36 - that “is” catabolized
-
Line 68 - Date not shown - it was supposed to be “data not shown”?
-
Line 152 - “acitivate”?
-
Line 156 - please rephrase for clarity
-
Line 166-167 - Please paraphrase this sentence - “maybe suggest” and ending sentence with “acts” sounds confusing
-
Line 172 - “nuclei” rather than “nucleases”
-
Line 242 - There is a spiral sign in the concentrations
Reviewer 2 Report
The paper entitled “1,5-Anhydro-D-fructose exhibits satiety effects via the activation of oxytocin neurons in the paraventricular nucleus” written by Nakata M. et al. reported a suggestion that 1,5-AF activates PVN oxytocin neurons to suppress feeding, indicating its potential as the energy storage monitoring messenger to hypothalamus for integrative regulation of energy metabolism.
Thus is a well-written paper containing interesting results which merit publication. For the benefit of the reader, however, a number of points need clarifying and certain statements require further justification. These are given below.
1) Do the authors have any explanation as to why the PVN oxytocin neurons respond and the SON oxytocin neurons do not?
2) Do the authors observe any sex differences regarding the results of this study?
3 I recommend performing double immunohistochemical staining for c-Fos and oxytocin to directly demonstrate that it is the oxytocin neurons that exhibit the cFos response.
4. Have any changes been observed in the 1,5-AF group, such as changes in the number of astroglial cells around oxytocin neurons in the PVN?
5. Is this feeding inhibition a direct response of the oxytocin neurons or is there a following projection from the oxytocin neurons to other neurons involved in feeding regulation?
6. Control experiments (tests of specificity) of the antibodies used should be described.
Round 2
Reviewer 1 Report
We thank the authors for addressing most of our comments, and especially for adding Figures 3C-G, 4B, and Supplementary Figure 1. It is certainly not surprising that non-OXT neurons would also be involved in satiety.
Is there a reason that Supplementary Figure 1 is not moved to the main figure? Are OXT KOs expected to already have higher baseline food intake than WT controls? Is the general increase in food intake/BW expected? Overall, I recommend this Supplementary Figure is moved to the main Figure 5.
There are still quite a number of typos and grammatical errors, some are listed below.
Add space between number and unit (e.g. 0.1 mg) - it is inconsistent throughout the manuscript
Line 52 - extra “t”
Line 68 - Date not shown - it was supposed to be “data not shown”?
Line 90 - “neuron activation” rather than “excitability of neuron”
Line 177- Should not end the sentence with “to act”
Line 215 - “paroral”?
Line 226 - grammar incorrect (“is activates”)
Line 350 - "Two-way ANOVA" (singular)
Author Response
- Is there a reason that Supplementary Figure 1 is not moved to the main figure? Are OXT KOs expected to already have higher baseline food intake than WT controls? Is the general increase in food intake/BW expected? Overall, I recommend this Supplementary Figure is moved to the main Figure 5.
Thank you for your valuable comments. We felt that data would be duplicated. However, in the latest current version, Supplemental Figure 1 has been moved to Figure 5B.
- There are still quite a number of typos and grammatical errors, some are listed below.
We apologize for our mistake. We have carefully corrected the manuscript again.
Reviewer 2 Report
The revised paper could be agreeable for publishing to the journal.
Author Response
Thank you for your valuable comments and agreement with the manuscript. We will further expand our research in the future.